# Research and Application of Multi-Mode Joint Monitoring System for Shaft Wall Deformation

**DOI:** 10.3390/s22176551

**Published:** 2022-08-30

**Authors:** Xinqiu Fang, Fan Zhang, Zongshen Shi, Minfu Liang, Yang Song

**Affiliations:** 1School of Mines, China University of Mining and Technology, Xuzhou 221116, China; 2Research Center of Intelligent Mining, China University of Mining and Technology, Xuzhou 221116, China; 3Xinji Energy Co., Ltd., China National Coal Group Corp., Bozhou 236000, China

**Keywords:** shaft wall rupture, the FBG, strain analysis, monitoring system

## Abstract

The mine shaft is an important channel linking the underground with the surface, undertaking important functions such as personnel and material transportation and ventilation. Thus the shaft, known as the throat of the mine, is the production hub of the whole mine. Since 1980, damage to coal mine shafts has occurred in many areas of China, which has seriously impacted the safety of mine production. Therefore, real-time monitoring of the shaft wall condition is necessary. However, the traditional monitoring method cannot achieve long-term, continuous and stable monitoring of the shaft wall due to the harsh production environment downhole. Hence, a multi-mode joint sensing system for shaft wall deformation and damage is proposed, which is mainly based on FBG sensing and supplemented by vibrating-string sensing. The principle of FBG sensing is that when the external environment such as temperature, pressure and strain changes, the characteristics of light transmission in the FBG such as wavelength, phase and amplitude will also change accordingly. Using the linear relationship between the strain and the wavelength shift of the FBG, the strain of the measured structure is obtained by calculation. Firstly, this paper introduces the basic situations of the mine and analyzes the causes shaft damage. Then the vertical and circumferential theoretical values at different shaft depths are derived in combination with the corresponding force characteristics. Moreover, a four-layer strain transfer structure model of the shaft consisting of the fiber, the protective layer, the bonding layer and the borehole wall is established, which leads to the derivation of the strain transfer relational expression for the surface-mounted FBG sensing on the shaft wall. The strain-sensing transfer law and the factors influencing the strain-sensing transfer of the surface-mounted FBG on the shaft wall are analyzed. The order of key factors influencing the strain-sensing transfer is obtained by numerical simulation: the radius of the protective layer, the length of the FBG paste, and the elastic modulus of the adhesive layer. The packaging parameters with the best strain-sensing transfer of the surface-mounted FBG on the shaft wall are determined. A total of six horizontal level monitoring stations are arranged in a coal mine auxiliary shaft. Through the comprehensive analysis of the sensing data of the two sensors, the results show that the average shaft wall strain–transfer efficiency measured by the FBG sensor reaches 94.02%. The relative average error with the theoretical derivation of shaft wall transfer efficiency (98.6%) is 4.65%, which verifies the strain transfer effect of the surface-mounted FBG applied to the shaft wall. The shaft wall’s deformation monitoring system with FBG sensing as the main and vibrating-string sensing as the supplement is important to realize the early warning of well-wall deformation and further research of the shaft wall rupture mechanism.

## 1. Introduction

According to the 2021 Coal Industry Development Annual Report, the nation’s raw coal production reached a record high of 4.13 billion tons in 2021. Moreover, coal consumption accounted for 56% of China’s total energy consumption. In the next half century, the energy structure dominated by coal will not change [1,2]. However, after a long period of high-intensity mining, coal resources buried at shallow depths have tended to deplete. Coal mining will continue to extend deeper. “To the underground to resources” has become the future development of China’s mining industry trends [3,4]. In the whole mining system, the shaft is the throat part of the mine. Since the 1980s, hundreds of shafts in China’s Huaibei, Huainan, Xuzhou and Yanzhou mines have been damaged to varying degrees [5], all of which have seriously impacted the safety of mine production. Therefore, the real-time monitoring of shaft deformation is not only a pivotal work to ensure safe production in mines, but also an important reference for shaft maintenance plans.

Traditional shaft wall monitoring methods mainly include GPS monitoring technology [6,7], the 3D deformation monitoring method [8,9], the inverted hammer method [10], the steel wire datum method [11] and the laser datum method [12]. However, owing to the complex natural environment and harsh measurement conditions downhole, the instruments for conventional measurements are subject to constant erosion. As a result, the traditional monitoring methods have low precision and cannot carry out long-term, continuous and stable monitoring. In summary, the traditional monitoring methods can no longer meet the needs of deepening mining for shaft wall safety monitoring.

In recent years, the FBG sensing technology has been increasingly applied by experts and scholars to monitor the status of bridges, ground settlements, dams and other projects because of its strong anti-interference ability, corrosion resistance and excellent perception [13,14,15]. Wang Weiming [16] et al. used the orthogonal test method to analyze the effects of shaft depth, the structure type of shaft wall and other factors on the stress distribution pattern of the shaft wall the intelligent test system to overcome a variety of harsh environments in the shaft to achieve intelligent monitoring of deep vertical shafts in coal mines. Liang Minfu [17] et al. established a substrate FBG strain-sensing transfer model and derived the sensing transfer factor equation by analyzing the forces. At the same time, the key factors affecting the transmission efficiency of FBG were explored to provide a theoretical basis for the design of surface-mounted FBG strain sensors. Chai Jing [18] et al. established the strain transfer model of the FBG and loose layer and deduced a new strain transfer equation. In addition, strain sensors were set up in 12 layers of the loose layer of the mine to explore and verify the characteristics of water loss settlement and deformation in the deep thick loose layer, which laid the foundation for the realization of shaft deformation monitoring in the future. Zhang Shubi [19] et al. proposed to combine the steel-wire datum method with distributed FBG sensing technology to monitor shaft wall strain and used experimental simulations to visualize the feasibility of the monitoring system.

This paper proposes a multi-mode joint monitoring system for shaft wall deformation, mainly based on FBG sensing supplemented by vibrating-string sensing. Based on the shaft wall’s deformation damage and force characteristics, a four-layer strain transfer structure model consisting of the fiber, the protective layer, the bonding layer and the borehole wall—is established. The strain transfer equation of the surface-mounted FBG on the shaft wall is derived, and factors such as the radius of the protective layer, the FBG paste length and the elastic modulus of the adhesive layer are obtained to have important effects on the strain transfer efficiency. Combined with MATLAB simulation, the optimal package parameters of the sensor are determined to make it possible to measure the shaft wall strain as accurately as possible. Finally, a field industrial test is conducted in a coal mine auxiliary shaft to investigate and verify the effect of the field application of the shaft wall deformation monitoring system with FBG sensing as the main and vibrating-string sensing as the supplement.

## 2. Analysis of Shaft Wall Damage Mechanism and Force Characteristics

### 2.1. Project Profile

A coal mine consists of three shafts: the main shaft, the auxiliary shaft and the air shaft. The topsoil layer of the shaft is constructed by the drilling method, and the bedrock layer is constructed by the drill-and-blast method. The monitoring site for this industrial test is the auxiliary well. A water inrush accident occurred in the auxiliary well, which suffered severe damage to itself and caused different degrees of damage to the other two wells. The basic situation of the auxiliary shaft of this coal mine: the elevation of the wellhead is +27.5 m, the elevation of the bottom of the well is −768 m, and the depth of the shaft is 795.5 m, of which 0~638.3 m is the topsoil section and 638.3~795.5 m is the bedrock section. The stratum crossed by the auxiliary shaft is the Cenozoic loose strata and the Permian. The Cenozoic loose strata are divided into four aquifers and three impermeable layers from the bottom up. Moreover, the fourth aquifer’s water abundance is moderate.

### 2.2. Mechanisms of Shaft Wall Damage

Since the 1990s, many experts and scholars in the field of mining engineering have studied the causes of shaft wall damage and proposed a series of shaft wall fracture theories. The current academic circle believes that damage to the shaft wall can be divided into two categories: non-mining damage and mining damage. The main hypotheses for the research of non-mining damage are as follows: ➀ “Neotectonic Movement Hypothesis” [20,21], which argues that neotectonic activity causes changes in subsurface stresses greater than the ultimate strain in the shaft wall, resulting in deformation and damage to the shaft. ➁ Substandard shaft design and construction quality cause damage to the shaft wall. However, based on statistical data, it is clear that many damaged shaft walls are repaired and then damaged again. Therefore, construction quality may be the cause of a small percentage of shaft wall damage, but it cannot be the primary cause of shaft damage. ➂ The “Vertical Additional Stress Hypothesis” [22,23] suggests that the water at the bottom of the deep topsoil layer is evacuated through the loose layer, resulting in a significant drop in water level. The soil in the upper layer thus settles and moves downward relative to the shaft, exerting a downward negative friction on the shaft wall. This force accumulates with depth and reaches a maximum at the junction of the deep topsoil layer and bedrock. When this stress exceeds the strain limit of the shaft, the shaft wall will rupture. This doctrine also rationalizes that most of the sites where shaft wall rupture occurs are concentrated near the interface between the topsoil layer and the bedrock. In addition to the above three main theories, there are “Seepage Deformation Hypothesis,” “Negative Friction Resistance Hypothesis” and “Three-factor Comprehensive Hypothesis” [24]. However, these hypotheses are not universally applicable because they can only explain the causes of damage in a small number of shafts, so they are not introduced in detail. Meanwhile, shaft wall rupture occurs mostly from April to October, especially in July and August. This feature indicates that temperature variation is also an essential factor affecting shaft rupture.

The reason for the damage to the coal mine borehole was that in the design of the underground building, the ingate and the cavern group of the auxiliary shaft were arranged in the mudstone stratum of poor rock strength at 730 m. As a result, the ingate and cavern group were deformed and damaged several times under earth pressure. After the accumulation of several destructive effects, the overlying strata was affected by repeated disturbances, and the stress state of the surrounding rock at the top of the ingate changed. The shaft wall was stretched by vertical stress and rupture occurred. Eventually, the fourth aquifer behind the shaft wall of this horizontal layer gushed into the shaft wall in a large amount. While the groundwater flooded into the shaft wall, it also carried a large amount of sediment, which damaged the lithology of the horizontal formation and caused the formation to sink. The subsidence of the strata further exerted additional vertical force on the shaft wall, which eventually resulted in severe damage and inclination of the auxiliary shaft [25]. The schematic diagram of wellbore damage is shown in Figure 1.

### 2.3. Analysis of Shaft Wall Force Characteristics

In the past, reinforced concrete was often used as the pouring material for coal mine boreholes. However, with the progress of construction technology, the multi-deck steel plate–concrete composite borehole wall is widely used in deep wells. In this paper, the tri-layer shaft wall is taken as the research object for force analysis, which is considered as an elastic composite cylinder. The schematic diagram of the shaft wall structure is shown in Figure 2. Here, the nth layer is selected for analysis.

(1)Horizontal lateral pressure


(1)
P=KH,


In the formula, *K* is the calculation coefficient, which ranges from 0.01 to 0.013; *H* is the depth of the selected position, *m*.

According to the formulas related to the elastic combination cylinder, the magnitude of the circumferential stress on the inner edge of the steel plate in the borehole wall under uniform horizontal pressure is:(2)$$\sigma_{\theta }=\frac{r_{1}^{2}t_{1}^{2}+r_{2}^{2}}{r_{1}^{2}t_{1}^{2}-r_{1}^{2}}P_{21},$$
where $P_{21}$ is the pressure between the concrete layer and the inner steel plate layer; $t_{1}$ is the ratio of the outer radius to the inner radius of the inner steel plate, $t_{1}=\frac{r_{2}}{r_{1}}$;
(3)$$P_{21}=\frac{k_{uv}^{\left(2\right)}}{k^{\left(1\right)}+k_{uu}^{\left(2\right)}}\times P,$$
where $k^{\left(1\right)}$ is the dislocation constant of the steel plate in the thin borehole wall, $k^{\left(1\right)}=\frac{{\left(r_{1}+r_{2}\right)}^{2}}{4hE}$; *h* is the thickness of the inner steel plate, m; *E* is the elastic modulus of the inner steel plate, $\frac{N}{m^{2}}$;

The dislocation constants of the concrete layers are calculated as follows:(4)$$k_{uv}^{\left(n\right)}=\frac{(1+\mu )r_{i}}{E}\cdot \frac{2t(1-\mu )}{t^{2}-1},\;k_{uu}^{\left(n\right)}=\frac{(1+\mu )r_{i}}{E}\cdot \frac{1+t^{2}-2\mu }{t^{2}-1},$$
(5)$$k_{vu}^{\left(n\right)}=\frac{(1+\mu )r_{o}}{E}\cdot \frac{2(1-\mu )}{t^{2}-1},\;k_{vv}^{\left(n\right)}=\frac{(1+\mu )r_{o}}{E}\cdot \frac{1+t^{2}-2\mu t^{2}}{t^{2}-1},$$
where $r_{i}$, $r_{o}$ are the radii of the inner and outer edges of the nth layer of the borehole wall, respectively, m; μ is the Poisson’s ratio of the borehole wall of the layer; *E* is the elastic modulus of the borehole wall of the layer, $\frac{N}{m^{2}}$;
(2)Self-gravity load
(6)$$\sigma_{z}=\gamma H,$$
where γ is the gravity density of borehole wall material. Reinforced concrete takes the value of 25 $kN/m^{3}$.

(3)Vertical additional stress

In the previous shaft wall design, only the effect of horizontal lateral pressure and self-gravity load on the borehole wall deformation was generally considered. However, with the in-depth research, temperature changes, soil subsidence caused by water loss of the aquifer at the bottom of the shaft wall due to mining and other factors will cause the superposition of additional vertical forces on the shaft wall. The current temperature stress generation can be considered from the following two aspects.

(1)The outer wall of the shaft wall is buried in a deep soil layer, and the temperature remains basically constant. The inner wall of the shaft wall is connected to the outside, thus causing a temperature difference between the inside and outside. At this time, an additional vertical stress will be generated at the outer edge of the shaft wall.(2)When the internal temperature of the shaft wall changes greatly, the borehole wall will deform in the radial and vertical directions due to the principle of thermal expansion and contraction. In the radial direction, the interaction force is generated due to the blocking of the surrounding deep soil layer. In the vertical direction, the expansion and contraction of the borehole wall cause the interaction between the borehole wall and the soil layer to generate negative friction.

Because the formation factors of additional vertical stress are complex, the proportion of each factor in the effect of additional stress has not yet been unified. Therefore, factors such as site construction and surrounding geological environment should be taken into account in the analysis of the deformation and rupture of the borehole wall.

From the above analysis of the relevant loads applied during the use of the shaft wall, we can calculate the expressions of the circumferential and vertical stresses at the inner edge of the shaft wall as follows:(7)$$\sigma_{\theta }=\frac{E}{1-\mu^{2}}(\varepsilon_{\theta }+\mu \varepsilon_{z}),$$
(8)$$\sigma_{z}=\frac{E}{1-\mu^{2}}(\varepsilon_{z}+\mu \varepsilon_{\theta }),$$
where *E* is the elastic modulus of the borehole wall; μ is the Poisson’s ratio of the borehole wall;
(9)$$\varepsilon_{\theta }=\frac{\sigma_{\theta }-\mu \sigma_{z}}{E},$$
(10)$$\varepsilon_{\theta }=\frac{\sigma_{z}-\mu \sigma_{\theta }}{E},$$

## 3. Strain-Sensing Transmission Mechanism of FBG on Shaft Wall Surface

### Analysis of Shaft Wall Force Characteristics

The FBG is formed by the characterization of the optical fiber by ultraviolet light to produce a periodic change in the axial refractive index to form phase gratings. Due to its structural changes, the grating can screen the incoming light signal. When the external environment such as temperature, pressure, strain, displacement and other factors change slightly, the characteristics of light transmission in the optical fiber such as wavelength, phase and amplitude will also change accordingly. Only light signals that meet a specific periodic variation can pass through the grating, while those that do not are reflected by the grating. In this way, the relationship between the two can be found through theoretical derivation and experimental verification to accurately measure the changes in the external environment. The Bragg wavelength that satisfies the reflection of the FBG is:(11)$$\lambda_{B}=2n_{eff}\Lambda ,$$

In the formula, $n_{eff}$ is effective refractive index of fiber core; Λ is grating period.

It can be obtained from Equation (11) that when the effective refractive index of the fiber core and the grating period change, the center wavelength of the reflected light will drift accordingly. Therefore, the FBG has the property of temperature–strain cross-sensitivity, which is extremely sensitive to both temperature and strain changes. Under the interaction of temperature and strain, the center wavelength drift of reflected light can be obtained as:(12)$$\frac{\Delta \lambda_{B}}{\lambda_{B}}=(1-P_{e})\varepsilon_{z}+(\alpha +\xi )\Delta T,$$

In the formula, α and ξ are the thermal expansion coefficient and thermo-optical coefficient of the FBG respectively; ΔT is temperature variation.

When using the FBG to monitor the borehole wall deformation, the bare FBG is relatively fragile and has poor shear resistance. If the bare FBG is directly pasted on the surface of the borehole wall, neither the transmission effect nor the service life can meet the requirements in practical application. Therefore, it is usually chosen to carve out grooves in the area where the stress is concentrated. After cleaning, the FBG sensor is laid in, and then the groove is covered with adhesive for pasting and packaging. In this way, the fiber structure can be better protected, and its service life can be improved. In addition, it can also increase the contact area between the FBG and the borehole wall to enhance the perception effect. The FBG sensor arrangement model diagram is shown in Figure 3.

When a surface-mounted FBG sensor approach is used, both the protective and adhesive layers between the fiber and the shaft wall absorb part of the energy during the strain transfer process. This causes the strain on the shaft wall to deviate considerably from the strain sensed by the FBG. Therefore, it is necessary to establish a four-layer strain transfer structure model of the shaft wall consisting of the fiber, the protective layer, the bonding layer and the borehole wall and mechanically analyze the model to derive the strain transfer equation adapted to the shaft wall: the FBG model.

The following assumptions are made for the strain transfer model: (1) The fiber, the protective layer, the bonding layer, and the borehole wall fit each other tightly, with a complete contact surface and no relative slippage. (2) In the whole strain transfer model, only the borehole wall is uniformly strained along the fiber axial direction, and radial strain is not considered. (3) Due to the complexity of the environment, the adhesive layer could not fit into the engraved groove. However, for the convenience of analysis and calculation, the ideal state is assumed here, so that $r_{m}=2r_{c}$.Analysis of the forces on each layer in the strain transfer model. The analysis of the forces on each layer in the strain model is shown in Figure 4.

Taking the optical fiber as the research object, the axial stress of the fiber is 0. From this, the equation is obtained as follows:(13)$$\pi r_{q}^{2}\left[\sigma_{q}\left(x\right)+d\sigma_{q}\left(x\right)\right]+2\pi r_{q}dx\cdot \tau_{cq}\left(x\right)-\pi r_{q}^{2}\sigma_{q}\left(x\right)=0,$$
where $\sigma_{q}$ is the axial stress of the fiber layer; $\tau_{cq}$ is the shear stress between the protective layer and the fiber layer;

Simplify Equation (13) as follows:(14)$$\tau_{cq}(x)=\frac{-r_{q}d\sigma_{q}(x)}{2dx},$$

Similarly, when the protective layer is taken as the research object, the force in the horizontal direction is also 0, and the equation can be simplified to obtain:(15)$$\tau_{mc}(x{,r}_{c})=\frac{-r_{q}^{2}}{2r_{c}}\frac{d\sigma_{q}(x)}{dx}-\frac{r_{c}^{2}-r_{q}^{2}}{2r_{c}}\frac{d\sigma_{c}(x)}{dx},$$
where $\sigma_{c}$ is the axial stress of the fiber layer; $\tau_{mc}$ is the shear stress between the protective layer and the fiber layer;

Assuming equal strain gradients across the layers of the FBG sensor:(16)$$\frac{d\varepsilon_{q}(x)}{dx}≅\frac{d\varepsilon_{c}(x)}{dx},$$

Substituting Equations (15) and (17) into Equation (16), it can be obtained:(17)$$\tau_{mc}(x{,r}_{c})=-\frac{r_{q}^{2}}{2r_{c}}[\frac{E_{q}d\varepsilon_{q}(x)}{dx}-\frac{\left(r_{c}^{2}-r_{q}^{2}\right)}{r_{q}^{2}}\frac{E_{c}d\varepsilon_{c}(x)}{dx}],$$

In the formula, $E_{q}$ is the elastic modulus of the fiber, Pa; $E_{c}$ is the elastic modulus of the protective layer, Pa.

Due to the difference of one order of magnitude between the elastic modulus of the fiber and the elastic modulus of the protective layer, it can be assumed that:(18)$$\frac{r_{c}^{2}-r_{q}^{2}}{r_{q}^{2}}\frac{E_{c}d\varepsilon_{c}(x)}{E_{q}dx}≅0$$

According to the above derivation and combined with the relevant physical relationship of materials, it can be obtained:(19)$$\tau_{mc}(x,r_{c})=-\frac{E_{q}r_{q}^{2}}{2r_{c}}\frac{d\varepsilon_{q}(x)}{dx}=G_{c}\frac{du_{c}(x)}{dy}$$

Taking the integrative of y for the two sides of Equation (19):(20)$$\int_{r_{q}}^{r_{c}}[-\frac{E_{q}r_{q}^{2}}{2r_{c}}\frac{d\varepsilon_{q}(x)}{dx}]dy=\int_{r_{q}}^{r_{c}}[G_{c}\frac{du_{c}(x)}{dy}]dy,$$
(21)$$u_{c}(x)-u_{q}(x)=-\frac{E_{q}r_{q}{}^{2}}{2G_{c}}\ln(\frac{r_{c}}{r_{q}})\frac{d\varepsilon_{q}(x)}{dx},$$

Through the above derivation, when the bonding layer is taken as the research object, it can also be deduced:(22)$$u_{g}(x)-u_{q}(x)=-\frac{E_{q}r_{q}{}^{2}}{2}\left[\frac{1}{G_{c}}\ln(\frac{r_{c}}{r_{q}})+\frac{1}{G_{m}}\ln(\frac{r_{m}{}^{\prime }}{r_{c}})\right]\frac{d\varepsilon_{q}(x)}{dx}$$

In the formula, $r_{m}^{\prime }=\frac{\sqrt{2}}{2}r_{m}$, $r_{m}=2r_{c}$, $G_{c}=\frac{E_{c}}{2(1+\mu_{c})}$, and $G_{m}=\frac{E_{m}}{2(1+\mu_{m})}$, substituting the above formulas into Equation (22) can obtain:(23)$$u_{g}(x)-u_{q}(x)=-E_{q}r_{q}{}^{2}\left[\frac{1+\mu_{c}}{E_{c}}\ln(\frac{r_{c}}{r_{q}})+\frac{1+\mu_{m}}{E_{m}}\ln\sqrt{2}\right]\frac{d\varepsilon_{q}(x)}{dx}$$
(24)$$k^{2}=\frac{1}{E_{q}r_{q}{}^{2}\left[\frac{1+\mu_{c}}{E_{c}}\ln(\frac{r_{c}}{r_{q}})+\frac{1+\mu_{m}}{E_{m}}\ln\sqrt{2}\right]}$$

In the formula, *k* is the perceptual lag factor;

Taking the derivative of *x* for the two sides of Equation (23):(25)$$\varepsilon_{q}(x)=\varepsilon_{g}(x)\left[1-\frac{\cosh(kx)}{\cosh(kL)}\right]$$

The average strain perception transfer rate between the shaft wall and the FBG can be expressed as:(26)$$\bar{\eta }(x)=\frac{\bar{\varepsilon_{q}}(x)}{\bar{\varepsilon_{g}}(x)}=\frac{2\int_{0}^{L}\varepsilon_{q}(x)dx}{2L\varepsilon_{g}(x)}=1-\frac{\sinh(kL)}{kL\cosh(kL)}=1-\frac{1}{kL}\tanh(kL)$$

## 4. Analysis of Strain-Sensing Transmission Factors between Shaft Wall and the FBG

In the practical application of surface-mounted FBG sensing, the average strain-sensing rate needs to be maximized to make the strain sensed by the FBG as consistent as possible with the strain of the measured object. According to Equations (25) and (26), the factors affecting the average strain perception transfer rate between the shaft wall and the FBG are the FBG paste length *L*, the radius of the protective layer $r_{c}$, and the elastic modulus of the adhesive layer $E_{m}$.

First of all, it is necessary to set relevant parameters, and then the factors affecting strain transfer are analyzed one by one. The radius of the FBG is $r_{q}=62.5\;\mu m$; elastic modulus is $E_{q}=7.2\times 10^{10}\;Pa$; Poisson’s ratio is 0.17 and the Poisson’s ratio of the protective layer is $\mu_{c}=0.3$.

It can be seen from the analysis in Figure 5 that when the radius of the protective layer is certain, the longer the length of the FBG pasted, the greater the average strain perception rate. However, the magnitude of change becomes slower and slower and finally smooths out, and the final transfer efficiency reaches more than 95%. When the length of the FBG pasted is small, the effect of the protective layer radius on the strain-sensing transmission is noticeable. From the slope of the curves in both directions, the influence of the protective layer radius on the average strain perception rate is more obvious than that of the FBG paste length. Therefore, when arranging the FBG sensor, the size of the groove should be considered first. Here, the FBG paste length is selected as 100 mm.

From Figure 6, we can intuitively see that as the radius of the protective layer continues to increase, the strain transfer efficiency becomes lower. According to the ideal situation set in the previous theoretical derivation, $r_{m}=2r_{c}$. Namely, the radius of the protective layer is larger, which means that the size of the groove engraved on the borehole wall is larger. This results in a thicker bonding layer, which prevents the FBG from fitting closely to the borehole wall and reduces the strain transfer rate. However, during the actual arrangement, the space in the borehole is narrow, and the construction is difficult. Thus, the groove size cannot be processed to the ideal size. To leave a margin for the construction process and to better protect the fiber to ensure a higher survival rate, a protective layer radius of 10 mm was chosen.

It can be seen from the analysis in Figure 7 that with the increase of the elastic modulus of the bonding layer, the average strain-sensing transfer factor also increases. However, it can be seen from the change in the numerical units of the vertical axis of the coordinate that the modulus of the bonding layer has little effect on the strain transfer effect. In addition, the smaller the Poisson’s ratio of the adhesive layer is, the better the strain transfer effect will be.

Combined with numerical simulations, the optimal parameters of the FBG-sensing encapsulation are determined by synthesizing the analysis of the degree of influence of each factor. The physical parameter values are shown in Table 1.

Substituting the above parameters into Equation (25), the relational expression of strain transfer between the FBG and the borehole wall can be calculated as follows:(27)$$\varepsilon_{q}(x)=0.986\varepsilon_{g}(x)$$

## 5. Field Industrial Test

### 5.1. Shaft Wall Monitoring System

#### 5.1.1. Monitoring System Principle

The shaft wall’s deformation monitoring system is divided into above and below ground. The downhole part senses the borehole wall strain by placing FBG sensors in a slot carved into the wall and then connecting the FBG sensors to fiber grating demodulators. The demodulator converts the sensor monitoring wavelength change signal into an electrical signal that is easier to recognize and process. Finally, the data are transmitted to the ground centralized control center for analysis and processing through cables. The theoretical limit value in case of rupture can be calculated from the analysis of the stress characteristics of the shaft wall in the previous section. This value is processed and set in the software. The system immediately alerts when the measured value is close to or exceeds the value. The construction personnel can quickly survey the location of stress abnormalities according to the monitored layer. At the same time, the data can be automatically recorded and stored at the terminal. In this way, the abundance data can be analyzed to obtain the shaft wall stress variation law and provide support for the continuous in-depth study of shaft wall deformation.

#### 5.1.2. Monitoring Methods

The underground monitoring part mainly adopts two types of sensors, namely vibrating-string sensors and FBG sensor, to achieve the monitoring of shaft wall deformation. The reason for using two methods of monitoring is that they can be used as a comparison for each other, and the data can be analyzed to compare which one is more accurate in measuring shaft wall strain. In addition, when the test verification is completed and put into practical application, the combination of FBG sensing as the main and the vibrating-string sensing monitoring as a supplement can be used to achieve accurate monitoring of shaft wall deformation. In this way, we give full play to the advantages of FBG sensing for accurate, long-term and stable monitoring of small deformations, but also to incorporate the advantages of vibrating-string monitoring for larger deformations occurring at specified depths in the shaft wall.

#### 5.1.3. Sensors Arrangement

Most of the locations in our test mine where the shaft deformation occurred are more than 100 m from the surface, and it was found that the repaired shaft would rupture repeatedly after long-term observation. At the same time, 638.3 m from the surface of the mine shaft is the junction of the topsoil section and the bedrock section. This is the place where the additional stress value is the largest. To summarize the above points, the test chooses to monitor the borehole wall strain in six horizontal layers at the depth of 300 m~600 m.

(1)Vibrating-string sensors arrangement

Four measurement points are uniformly arranged on the inner side of the shaft wall at a horizontal level, with each of the two measurement points at 90°. Each measurement point is placed with one vibrating-string sensor in the vertical and circular directions, respectively.

(2)FBG sensors arrangement

The sensors of each horizontal layer are equally spaced on the inner side of the shaft wall, and the location arrangement is basically the same as that of the vibrating-string sensor. Thus, a total of 10 FBG strain sensors are installed on each horizontal level. The FBG sensors at each level are connected to each other in series. The FBG used in the sensor in the field test is characterized by the optical fiber of type SMF-28E. The center wavelength of the FBG sensor used in each horizontal level is 1530.756 nm, 1535.858 nm, 1541.019 nm and 1545.752 nm. After all sensors are installed, a total of seven channels in three horizontal layers are connected to the main optical cable. The sensor installation process first selects where the shaft wall strain sensor is arranged. The second step is to use a drilling rig to make grooves on the surface of the shaft wall according to a predetermined size. After that, the surface is polished and cleaned, and the packaged FBG sensor is placed. Epoxy is injected into the groove to stick the sensor in place. Finally, bolts are used to fix the fiber grating sensor further. Since the FBG has the characteristics of easy to break and poor shear resistance, bending should be minimized to avoid damage to the sensor during the sensor installation process. At the same time, waterproof treatment should be performed. After the sensor installation, the excess fiber is coiled and fixed with straps. A protective cover is added to the sensor to further improve its survival rate.

Factors such as thermal effects and residual strain during sensor installation will cause large errors in monitoring accuracy, so this problem needs to be discussed and overcome. Before installation, the FBG sensors are connected to the FBG demodulator for wavelength testing and recording. After each horizontal level’s sensor installation, the sensor’s wavelength test is tested again. Therefore, the center wavelength value of the fiber grating sensor in the system is zero-calibrated so that the data obtained by subsequent monitoring is the real strain value of the shaft wall.

The FBGs have temperature–strain cross-sensitivity properties. Therefore, to avoid the influence of temperature changes on the measurement accuracy of the FBG sensors, the common temperature compensation methods include the double parameter matrix method, the unstressed grating temperature compensation method, the polymer encapsulation method and the negative temperature material method. In this paper, the unstressed grating compensation method is adopted. Two FBG sensors are placed on opposite sides of each FBG monitoring level to measure temperature changes. We subtract the wavelength change of the compensation sensor from the center wavelength change of the working strain sensor to obtain the wavelength change caused by the strain of the shaft wall and achieve temperature strain compensation. The FBG and vibrating-string sensor arrangement of each layer is shown in Table 2.

The shaft wall monitoring sensor arrangement is shown in Figure 8.

#### 5.1.4. The FBG Sensing Demodulation System

The FBG demodulation system is an essential part of the FBG sensing monitoring system. The grating wavelength demodulation technology can demodulate the spectral center wavelength variation into an intuitive physical parameter variation. Therefore, adopting the FBG demodulation system with fast demodulation speed and high precision in field industrial tests is necessary. The workflow is that the light emitted by the broadband light source enters the optical fiber through the coupler. Then, the light source is time divisionally incident on the FBG sensing channel through the optical switch to scan the FBG. The reflected light of the FBG sensors in each channel is time divisionally entered into the coupler through the optical switch. At this time, the stress change of the shaft wall is converted into optical pulse signals. After that, the optical pulse signals are converted into electrical pulse signals by the demodulator. Finally, after A/D conversion, it becomes digital signals and enters the data processing unit to complete the wavelength demodulation. In the sensing optical path and system circuit, the noise will inevitably be generated. Therefore, in the data processing unit, digital filtering and an improved peak-seeking algorithm will be used to improve the demodulation accuracy. However, under the effect of non-uniform transverse stress, the non-uniform strain will distort the wavelength of the FBG. Thus, a layer-peeling algorithm can be introduced into the data processing unit to calculate the complex coupling coefficient of each layered grating. In this way, the demodulation of the non-uniform strain of the FBG is realized.

The FBG sensing demodulation system is shown in Figure 9.

### 5.2. Analysis of Monitoring Data

To compare the sensing effects of the FBG sensor and the vibrating wire sensor more intuitively, the horizontal levels at 579 m and 581 m are selected for analysis and comparison. Firstly, by analyzing the force characteristics of the borehole wall in Section 2.2, it can be calculated that the vertical strain of the borehole wall at 579 m is 85.6198 με and at 581 m is 87.6342 με. Figure 10 and Figure 11 present the graphs of FBG sensing and vibrating wire sensing to monitor the strain variation of the borehole wall, respectively.

As can be seen from Figure 10, the maximum vertical compressive strain measured by the FBG sensor at 579 m is −79.199 με, the maximum vertical tensile strain is 194.683 με, and the vertical average strain is 80.657 με.

As can be seen from Figure 11, the maximum vertical compressive strain measured by the vibrating wire sensor at 581 m is −304.7044555 με, the maximum vertical tensile strain is 73.49 με, and the vertical average strain is −105.91 με.

Comparing the data obtained by the two sensors shows that the FBG sensing can achieve long-term and stable monitoring to obtain abundant borehole deformation data. The results show that the measured borehole wall strain transfer efficiency of the FBG sensor reaches 92.1%, and the relative error with the theoretically deduced borehole wall transfer efficiency (98.6%) is 6.6%.

At the same time, the horizontal levels at 396 m and 640 m are selected to further verify the reliability and accuracy of the FBG sensor monitoring. Firstly, it can be calculated that the vertical strain of the shaft wall at 396 m is 72.484 με and at 640 m is 103.1506 με. Figure 12 and Figure 13 present the graphs of FBG sensing to monitor the strain variation of the shaft wall at 396 m and 640 m, respectively.

As can be seen from Figure 11, the maximum vertical compressive strain measured by the FBG sensor at 396 m is −54.088 με, the maximum vertical tensile strain is 222.774 με, and the vertical average strain is 67.3975 με. The results show that the measured shaft wall strain transfer efficiency of the FBG sensor reaches 92.98%, and the relative error with the theoretically deduced borehole wall transfer efficiency (98.6%) is 5.7%.

As can be seen from Figure 12, the maximum vertical compressive strain measured by the FBG sensor at 396 m is −2.192 με, the maximum vertical tensile strain is 202.152 με, and the vertical average strain is 100.0443 με. The results show that the measured shaft wall strain transfer efficiency of the FBG sensor reaches 96.98%, and the relative error with the theoretically deduced borehole wall transfer efficiency (98.6%) is 1.64%.

In the long-term, the relationship between the measured borehole wall strain values from the FBG and the theoretical strain values of this horizontal layer is roughly under the relationship Equation (27). The results show that the average shaft wall strain transfer efficiency measured by the FBG sensor reaches 94.02%. The relative average error with the theoretical derivation of shaft wall transfer efficiency (98.6%) is 4.65%. At the same time, it is verified that the comprehensive monitoring system of FBG and vibrating-string sensing can realize real-time monitoring of key layers on the shaft wall. It provides data support for the ground personnel to analyze and evaluate the shaft wall status and further research on the rupture mechanism of the shaft wall fracture in the future.

## 6. Conclusions

(1)In this paper, the strain transfer structure model of a shaft wall FBG is established with surface-mounted FBG sensing as the research object. The relational expression of strain transfer between the FBG and the borehole wall is derived, and the optimal average strain-sensing transfer rate is obtained.(2)Through the analysis of the strain transfer equation and the numerical simulation with MATLAB, the factors that affect the average strain perception transfer rate between the borehole wall and the FBG include the FBG adhesion length, the radius of the protective layer, the elasticity modulus of the adhesive layer and so on. Among them, the radius of the protective layer has the most obvious effect on the average strain transfer perception factor, followed by the length of the FBG, while the elasticity modulus of the adhesive layer has the least effect on the strain transfer effect. In summary of the above analysis and combined with the actual construction technology and conditions, the physical parameters of the above influencing factors are taken as the optimal values to achieve the best strain transfer effect.(3)In the field industrial test, a shaft wall’s deformation monitoring system based on combining the new FBG sensing monitoring with the traditional vibrating wire sensing was arranged in the auxiliary shaft of the coal mine. By selecting the vibrating wire sensing and FBG sensing monitoring data of adjacent horizontal horizons for analysis, it was found that the strain value measured by the FBG sensor and the theoretical strain value calculated at the corresponding horizontal conform to the deduced relational formula. This shows that the shaft wall deformation system achieved the expected monitoring effect and can realize long-term real-time and reliable monitoring. In the future, more subsystems can be added to this monitoring system, such as the variation of groundwater level, the amount of ground subsidence, and the deformation of the tank can be used as a reference to predict the borehole wall deformation. This will provide more abundant and comprehensive data support for early warning of shaft wall deformation.

## Figures and Tables

**Figure 1 sensors-22-06551-f001:**
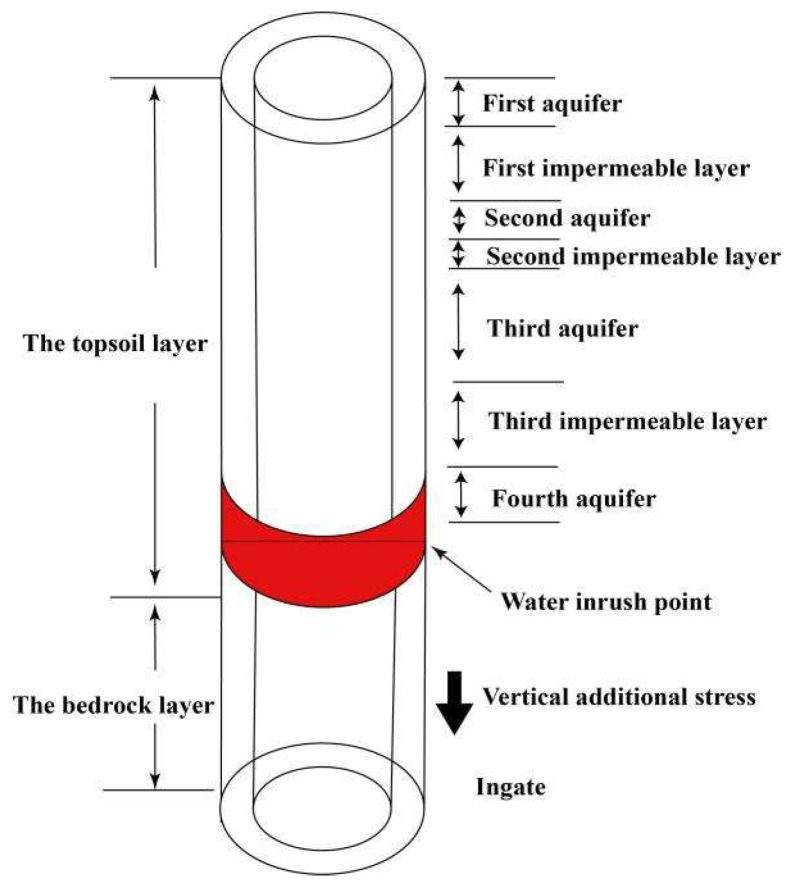
The schematic diagram of wellbore damage.

**Figure 2 sensors-22-06551-f002:**
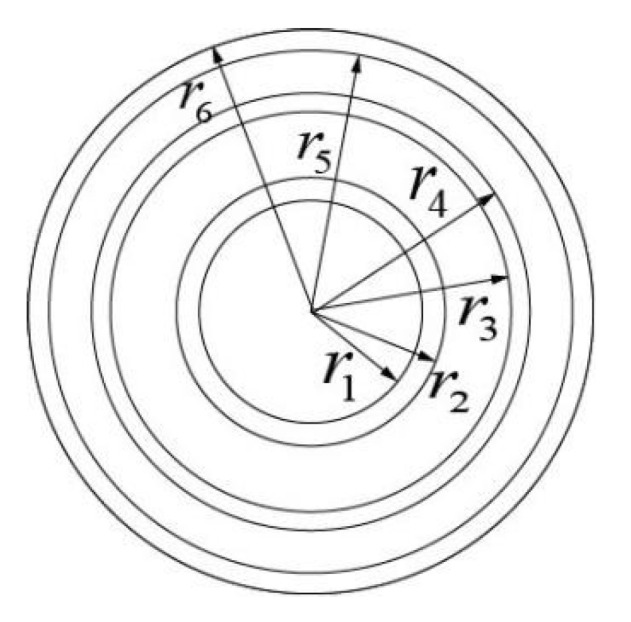
The schematic diagram of the shaft wall structure.

**Figure 3 sensors-22-06551-f003:**
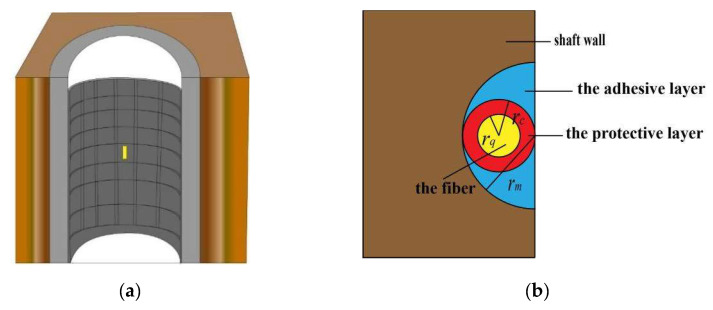
The FBG sensor arrangement model diagram: (**a**) cross-sectional view; (**b**) plane-form.

**Figure 4 sensors-22-06551-f004:**
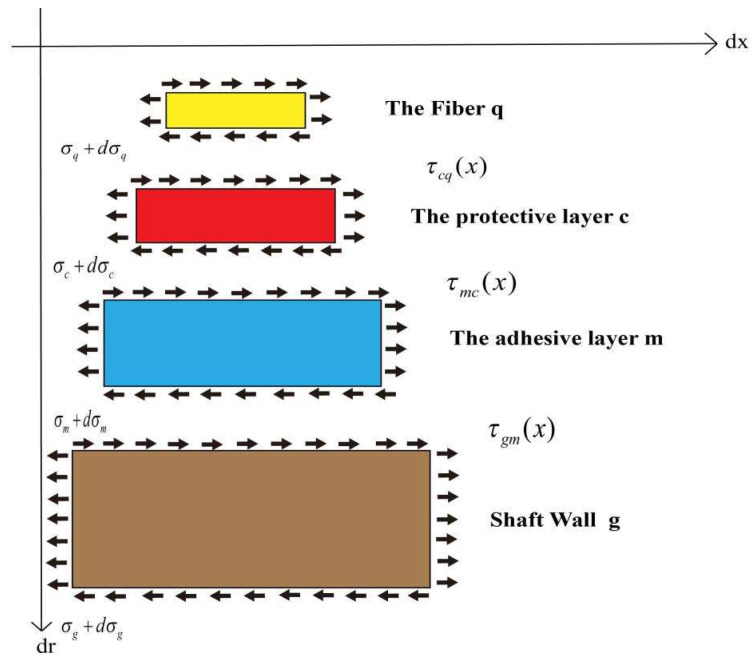
Analysis of the forces on each layer in the strain transfer model.

**Figure 5 sensors-22-06551-f005:**
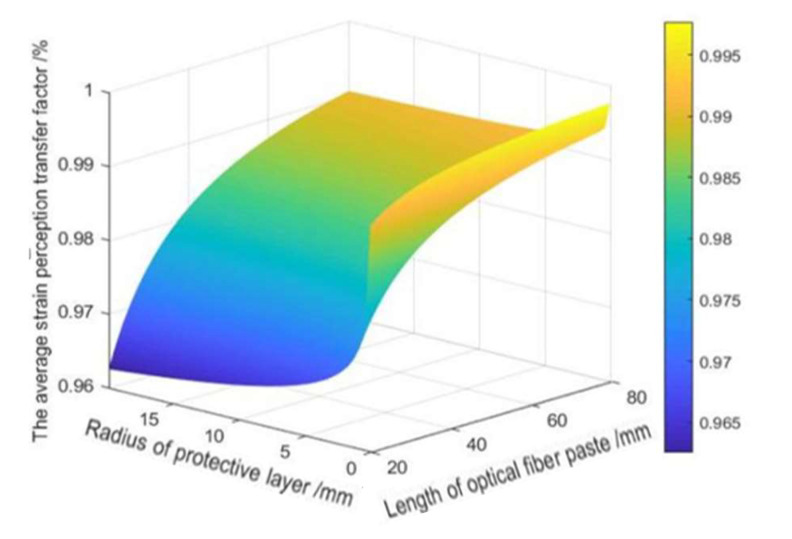
Effect of the length of FBG pasted and radius of the protective layer on an average strain perception transfer factor.

**Figure 6 sensors-22-06551-f006:**
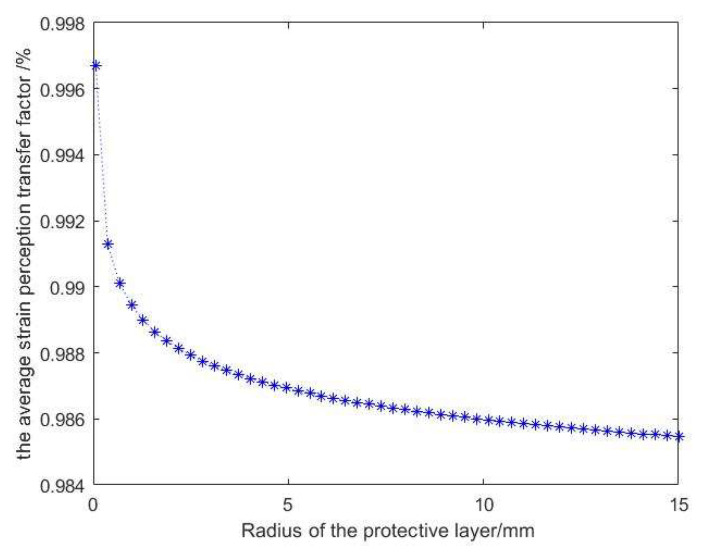
Effect of the radius of the protective layer on the average strain perception transfer factor.

**Figure 7 sensors-22-06551-f007:**
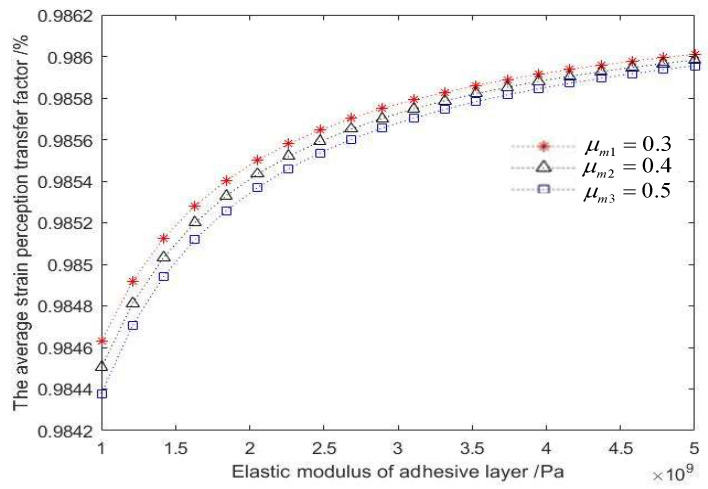
Effect of the adhesive layer elastic modulus on the average strain perception transfer factor.

**Figure 8 sensors-22-06551-f008:**
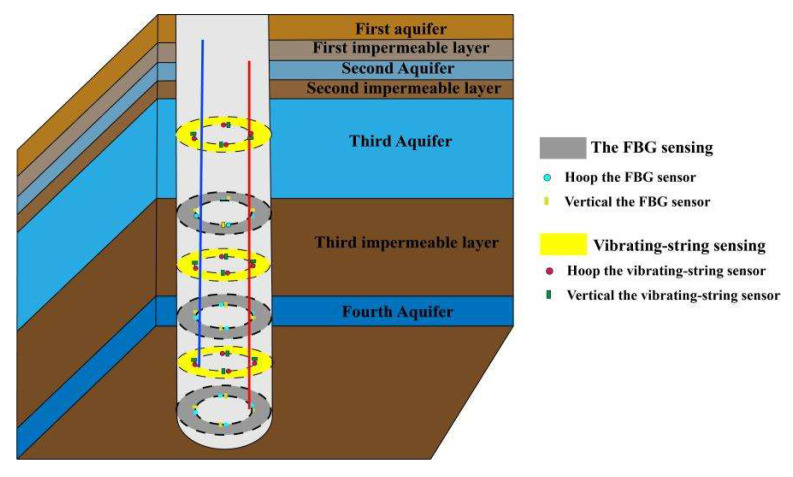
Arrangement of shaft wall monitoring sensors.

**Figure 9 sensors-22-06551-f009:**
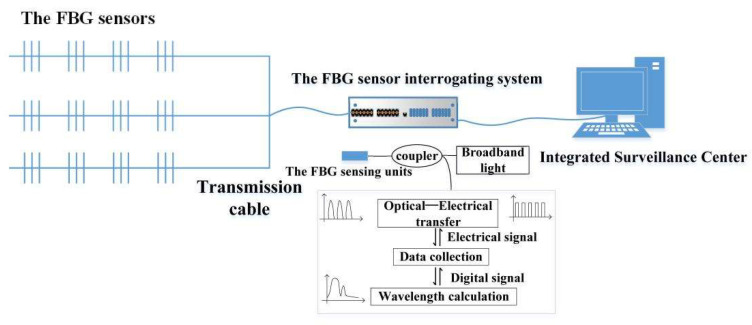
The FBG sensing demodulation system.

**Figure 10 sensors-22-06551-f010:**
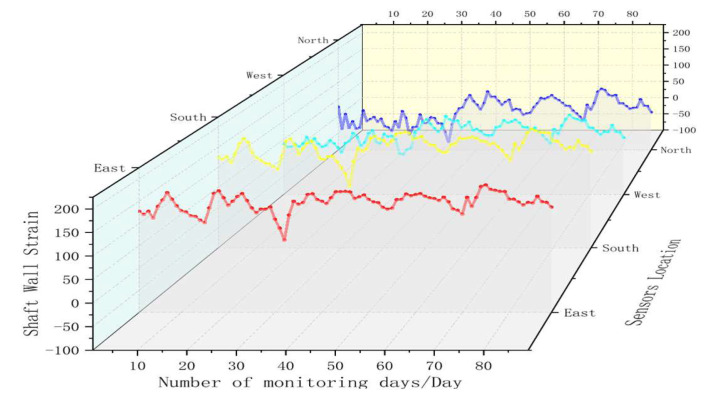
The FBG sensing strain diagram (579 m).

**Figure 11 sensors-22-06551-f011:**
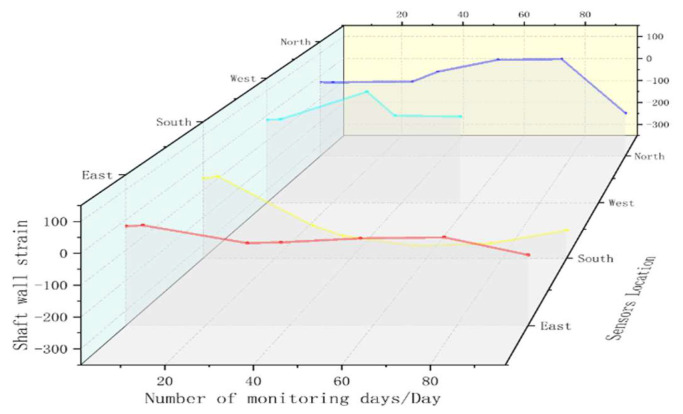
Vibrating-string sensing strain diagram (581 m).

**Figure 12 sensors-22-06551-f012:**
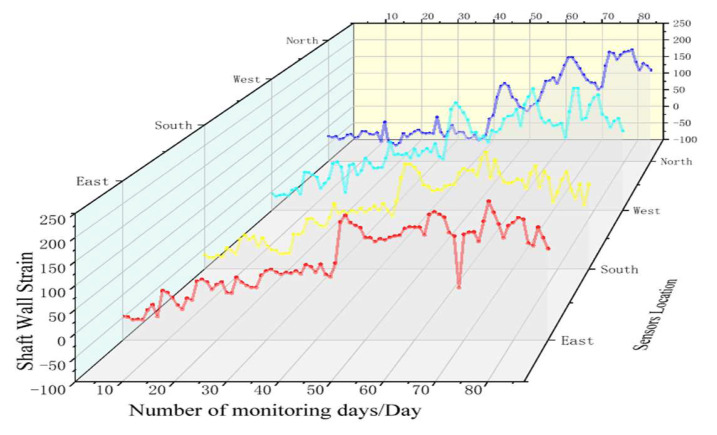
The FBG sensing strain diagram (396 m).

**Figure 13 sensors-22-06551-f013:**
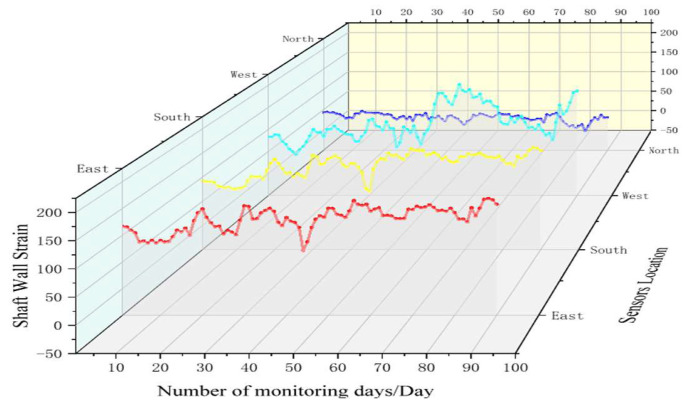
The FBG sensing strain diagram (640 m).

**Table 1 sensors-22-06551-t001:** Physical parameters of the strain-sensing transfer model.

Physical Parameters	Numerical Value	Unit
Length of optical fiber paste 2 L	100	mm
Radius of protective layer $r_{c}$	10	mm
Elastic modulus of protective layer $E_{c}$	4 × 10^9^	Pa
Radius of adhesive layer $r_{m}$	20	mm
Elastic modulus of adhesive layer $E_{m}$	5 × 10^9^	Pa
Poisson’s ratio of adhesive layer $\mu_{m}$	0.35	—

**Table 2 sensors-22-06551-t002:** FBG and vibrating-string sensor arrangement of each layer.

Number	Monitoring Horizontal Depth/m	Geotechnical Properties	Layer Thickness	Sensor Type	Number of Sensors
1	325 m	Fine medium sand	28.2 m	Vibrating-string sensor	8
2	396 m	Fine sand	18.4 m	FBG sensor	10
3	473 m	medium sand	7.6 m	Vibrating-string sensor	8
4	579 m	Fine sand	5. 86 m	FBG sensor	10
5	581 m	Fine sand	5.73 m	Vibrating-string sensor	8
6	640 m	siltstone	3.62 m	FBG sensor	10

## Data Availability

All data and code used or analyzed in this study are available from the corresponding author on reasonable request.

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
