# Peer review of "Research and Application of Multi-Mode Joint Monitoring System for Shaft Wall Deformation"

_sensors, 2022, doi:10.3390/s22176551_

Round 1

Reviewer 1 Report

This paper introduces a joint multimodal sensing system for well wall deformation damage with FBG sensing as the mainstay and vibrating string sensors as a supplement, which is of great significance for realising early warning of well wall deformation and further research on well wall damage mechanisms. The article is adequate, well-structured and well thought out, but the following issues remain to be revised as suggested.

The following are the questions and some mistakes in this manuscript:

(1)  In line 8 of the abstract, there is a reference to "FBG sensors", for which a brief comment is suggested.

(2)  The annotated text of the figures in the text is too small and the figures are generally blurred.

(3)  Equation (2) appears with t1 and no comment is given in the text.

(4)  Suggested supplementary diagram for the arrangement of vibrating string sensors.

(5)  In the monitoring data comparison section in section 5.2 on page 11, only the data from the 549m and 581m height sections were selected for comparison, and thus the reliability of this monitoring system was obtained, and it is recommended that more sets of data be compared.

(6)  The article is not very closely linked to the previous article, which describes many different causes of well wall damage, and the monitoring system described in the later article does not give a specific cause of well wall damage.

To conclude, this paper is well organized, but there are still many of the above problems that need to be revised.

Reviewer 2 Report

The paper proposed to monitor the integrity of mine shaft wall through the use of FBG and vibrating string sensor. A large part of the work is on the theoretical analysis about the stress transfer efficiency to the FBG. The remaining is an experimental demonstration of the FBG and vibrating string monitoring system in an actual mine shaft. The topic is useful for improving coal mine safety and in this respect, it is well worth publishing. However, a number of points have been overlooked in the experimental measurement as pointed out below. These should be thoroughly considered, corrected and discussed.

1. Sources for properties such as moduli, Poisson's ratios of the adhesive, protective layer and fiber employed for calculation should be duly referenced.

2. Illustrative fonts in Figure 7 are too small to be legible. Non English characters should be translated into English.

3. It is too vague to simply mention "The demodulator converts the sensor monitoring wavelength change signal into an electrical signal". Details of the interrogative technique for the FBG should be described. The demodulated electrical signal may not accurately reflects the strain for reason stated below!

4. From the stress analysis, it is clear that the shaft wall suffers longitudinal as well as circumferential stresses and the longitudinal stress is non-uniform in nature. It is well known that non-uniform stress along the FBG will chirp and broaden the characteristic Bragg waveform, which means that the λB in eqn. 11 is no longer a single value but a series of values. Moreover, transverse stress normal to the fiber will cause splitting of the Bragg waveform into two peaks. As a result, the FBG waveform will become a complex shape instead of single narrow peaked. In that case, how can the strain be obtained from the complex waveform?

5. The x-axes of Figures 8 and 9 are very confusing. What is the meaning of number of monitoring days? Is the data at say 60 day  is the one measured on the 60th day, or is it the average of 60days?

6. The strains reported by the vibrating string sensor and the FBG differ by quite a large percentage. Also their trends of variation do not match. Considering the corresponding sensors are installed quite near to each other, how can  the large discrepancies be explained?

7. As is evident from eqn.12, the shift in Bragg waveform is affected by temperature as well. Normally the peak wavelength will shift by 1pm with 1 με  and ~10pm with 1 oC. In other words, if the temperature effect has not been somehow removed, 1oC changes can lead to 10 μεapparent strain and the observed strain variation may be attributed to about 10 oC changes.  Differential thermal expansion of the protective layer and adhesive etc, may confer additional apparent strain. Figures 8 shows data spanning a number of days during which there will certainly be change in temperature. The paper seems to have overlooked this phenomenon. How much is the possible temperature change and how much variation in the FBG indicated strain be attributed to this temperature variation?

8. Although it has not been clearly stated, it appears that the embedment of the FBG was done after the borehole wall was in place. In that case, the strain that initially exists in the borehole wall will not be able to be reflected. Figures 8 and 9 both show that considerable initial strains was indicated at day 0. These must be residual strain generated by the sensor installation process. Subsequently measured strain will include these residual strains and thermally induced one. Any change in the borehole wall strain will be reflected to some degree in the indicated strain but the actual wall strain can only be obtained after correcting for initial wall strain before embedment, thermal effect and the residual strain due to sensor installation. The numbers reported in the in the text and the figures are therefore not the strain the borehole wall.
This should be pointed out, discussed and ways to overcome it should be devised for the monitoring system to be useful.

Reviewer 3 Report

In this contribution, the authors describe in detail an application for strain sensing related to mine shaft deformation and damage. The system makes use of FBGs mounted inside particular grooves created in the shaft wall. Since the FBGs are embedded in protective and adhesive layers, the author proposed a model to describe the strain transfer from the shaft wall to the FBGs. The authors provide a set of in-situ experiment that justify the FBGs system validity. The work is interesting, as well as the application of the FBGs system. The problem is carefully described, and the scientific and technical background is properly explained. The strain transfer theory is well explained and understood. The results are acceptable. However, I need to indicate some important improvements before to suggest the work for publication:

1) I suggest an extensive revision of English language. There are many sentences in the text with weird grammar construction. I suggest a good proofread from a native speaker.

2) I would like to see, described in the text, more pieces of information regarding the FBGs that have been used.

 - Are the FBGs inscribed inside standard SMF-28 telecom fiber?

 - If yes, why the fiber does not present the usual acrylate coating?

 - Are the FBGs custom made or bought from a manufacturer?

 - Are the FBGs arranged in an array on a single fiber?

 - If yes, can I know the values of the Bragg wavelengths?

3)   The author must describe the system used to interrogate the FBGs. In particular:

 - What is the source of the broadband light?

 - Which detector has been used to pick up the reflected wavelength shift?

4) The FBGs can detect strain and temperature, the Bragg wavelength shift depends on both these physical properties. How the authors discriminate temperature and strain in their experiment?

5) The authors must give more technical details regarding the protective layer material and the adhesive used to deploy the FBGs inside the wall grooves. 

6) Even if the application is interesting, I have concerns about the novelty. What is described is a quite standard FBGs application for boreholes and wells. The authors need to better explain the novelty of their work. 

Round 2

Reviewer 1 Report

This paper introduces a joint multimodal sensing system for well wall deformation damage with FBG sensing as the mainstay and vibrating string sensors as a supplement, which is of great significance for realising early warning of well wall deformation and further research on well wall damage mechanisms. The article is adequate, well-structured and well thought out, and has been improved relative to the previous one, but the following issues remain to be revised as suggested.

The following are the questions and some mistakes in this manuscript:

(1)   The figures are clearer than in the previous draft, the axis labels below Figure 4 are recommended to be aligned with the axes, and the “Um” for Poisson's ratio in Figure 6 is recommended to be aligned with the symbol for Poisson's ratio in the text.

(2)   In section 2.2 of the article the author talks about a number of hypotheses for well wall damage, mentioning “topsoil layers”, “ the bedrock” and other soil layers, but the reader is not aware of the exact location of these well wall damages, so it is suggested that a figure be added to help illustrate this.

(3)   It can be seen from Figure 7 that the FBG sensor and the vibrating string sensor are not installed at the same time in the same soil section and how a comparison between the FBG sensor and the vibrating string sensor can be carried out.

(4)   In line 439, the FBG sensing strain should be seen in Fig. 9, but it is written as Fig. 8; in line 444, the same problem is found, as the FBG sensing strain should be seen in Fig. 10, but it is written as Fig. 9.

(5)   The authors compare FBG sensing strain data from 396 m and 640 m horizontal layers, but do not compare vibrating string sensors data or give the strain transfer efficiency of the FBG sensors at the well wall, it is recommended that this piece be added and also that the response be revised in the conclusion and abstract.

To conclude, this paper is well organized.

Reviewer 2 Report

No further comment.

Author Response

Dear Reviewer #2:

    Thank you very much for your valuable comments on our paper. Those comments are very beneficial and all valuable for revising and improving our paper.

    Once again, thanks for your comments and suggestions.

Reviewer 3 Report

The authors improved the manuscript. Most of my concerns have been addressed, except for one/two.

1) Since the authors uses FBGs arrays for their system, I asked to include the specification of the FBGs in their manuscript. It is not a huge work to do, just to include the number of FBGs in the array, the Bragg wavelengths, which kind of fiber has been used for inscribing the FBGs (SMF28 I suppose), and the manufacturer if it is not problematic to include its name.

2) I asked if the FBGs array fiber presents or not the coating. Usually manufacturer removes the polymer coating to inscribe the array and then the fiber is re-coated. It is very unlikely that the manufacturer delivers the fiber without coating because the bare fiber is fragile. If the fiber, as I suspect, presents the polymer coating (radius of 125 um), the strain model of section 3.1 should be updated. So my question is: "Is the model of 3.1 valid even neglecting the presence of polymer coating around the fiber?" 
